# Impact of Sarcopenia, Dynapenia, and Obesity on Muscle Strength and Quality in Chronic Kidney Disease Patients: A Sex-Specific Study

**DOI:** 10.3390/healthcare13131621

**Published:** 2025-07-07

**Authors:** Marcio Bacci, Fernanda Rico Angelotto, Thiago Dos Santos Rosa, Thaís Branquinho De Araújo, Hugo De Luca Corrêa, Lysleine Alves De Deus, Rodrigo Vanerson Passos Neves, Andrea Lucena Reis, Rafael Lavarini dos Santos, Jéssica Mycaelle Da Silva Barbosa, Vitória Marra Da Motta Vilalva Mestrinho, Carmen Tzanno-Martins, Whitley J. Stone, Ivo Vieira De Sousa Neto, Wilson Max Almeida Monteiro de Moraes, Guilherme Borges Pereira, Jonato Prestes

**Affiliations:** 1Department of Physical Education, Catholic University of Brasilia, Brasilia 71966-700, Brazil; marciobacci@outlook.com (M.B.); thiagoacsdkp@yahoo.com.br (T.D.S.R.); thaisbranquinhodearaujo@gmail.com (T.B.D.A.); hugo.efucb@gmail.com (H.D.L.C.); lysleine.deus@a.ucb.br (L.A.D.D.); rpassosneves@yahoo.com.br (R.V.P.N.); andrealucereis@gmail.com (A.L.R.); lavarini.santos@gmail.com (R.L.d.S.); jessicamycaelle.nut@gmail.com (J.M.D.S.B.); vitoria.mestrinho@a.ucb.br (V.M.D.M.V.M.); wmaxnutri@gmail.com (W.M.A.M.d.M.); 2Laboratory of Clinical Exercise Physiology, Department of Physiological Sciences, Federal University of São Carlos, São Carlos 13565-905, Brazil; fernandara@estudante.ufscar.br (F.R.A.); gbp.ufscar@gmail.com (G.B.P.); 3NephroClinics, Premium Nephrology Clinic, Brasilia 70390-907, Brazil; 4Clinical Group Home Dialysis Center and RenalClass, São Paulo 01239-040, Brazil; gugudeluca@hotmail.com; 5School of Kinesiology Recreation and Sport, Western Kentucky University, Bowling Green, KY 42101, USA; whitley.stone@wku.edu; 6School of Physical Education and Sport of Ribeirao Preto, University of São Paulo (USP), São Paulo 05508-220, Brazil; ivoneto04@hotmail.com; 7Graduation Program on Physical Education, Catholic University of Brasilia, Q.S. 07, Lote 01, EPTC—Bloco G, Brasilia 71966-70, Brazil

**Keywords:** geriatric assessment, muscular atrophy, renal dialysis, muscle strength, aging

## Abstract

Sex-specific differences in the prevalence of sarcopenia, dynapenia, and the impact of obesity on muscle strength and quality in patients with chronic kidney disease (CKD) remain underexplored. **Background/Objectives**: In this cross-sectional study, 78 adults with stage 5 CKD undergoing thrice-weekly maintenance hemodialysis in Brazil (44 men, 34 women; mean ± SD age = 57.55 ± 4.06 years) were assessed. Anthropometry (BMI, waist circumference, waist-to-height ratio), dual-energy X-ray absorptiometry, circulating IL-6, Timed Up and Go, handgrip strength (Jamar ^®^ dynamometer), and muscle quality index (MQI = handgrip/BMI) were obtained. Dynapenia (handgrip < 27 kg men and < 16 kg women) and sarcopenia (1.0 kg/kg for men and 0.56 kg/kg for women) were classified using EWGSOP2-2018 and FNIH thresholds. **Results:** Compared with reference values, men showed markedly reduced muscle strength and muscle quality (men: handgrip 21.3 ± 5.1 kg; MQI 0.80 ± 0.23 AU) than women. Also, men were 5.1 times more likely to present with dynapenia (88.6%; 95% CI 2.28–11.60) and 3.15 times more likely to present with sarcopenia (75.0%; 95% CI 1.88–5.30) than women. Waist circumference, waist-to-height ratio, BMI, and body fat % correlated inversely with MQI in both sexes (*p* ≤ 0.01) and with handgrip strength in men (*p* ≤ 0.01) but not in women. **Conclusions:** Among hemodialysis patients, men exhibit a substantially higher burden of dynapenia and sarcopenia than women and excess adiposity is independently associated with poorer muscle quality in both sexes. These findings highlight the need for sex-specific screening cut-offs and integrated strategies targeting both muscle dysfunction and central obesity in CKD management.

## 1. Introduction

Chronic kidney disease (CKD) represents a definitive change in the function of the kidney, is irreversible, slow, and displays progressive evolution [1,2]. Unfortunately, the incidence and prevalence worldwide are increasing with poor outcomes and high costs [2]. In Brazil, the Brazilian Society of Nephrology revealed in 2016 that the prevalence and incidence of patients on dialysis continue to rise, and considering that CKD has an essential impact on the morbidity and mortality of patients [3], interventions aiming to lessen the occurrence of complications are welcome, as they might positively impact the prognosis of the affected patient.

One of the main complications verified in CKD patients is sarcopenia, and CKD has often been called a model of premature ageing [4,5]. Sarcopenia was first defined as loss of muscle mass by Irwin Rosenberg [6]. Sarcopenia in CKD is influenced by the confluence of uremia (metabolic acidosis and malnutrition), different diseases (diabetes, hypertension, and malignancy), immune system/inflammation (increases in inflammatory cytokines), hormones (hypogonadotropic hypogonadism and suppression of the pituitary–testicular axis), mechanical factors (physical inactivity), and a renin–angiotensin-activated system (increased angiotensin-2 levels) [7].

Although sarcopenia is an independent disease coded as M62.84 in the International Classification of Disesase-10 system in 2016, there is still a lack of standardized diagnostic criteria, as different organizations such as the European Working Group on Sarcopenia in Older People (EWGSOP-2010), Foundation for the National Institutes of Health (FNIH) Sarcopenia Project, and South Asian Working Action Group on SARCOpenia (SWAG-SARCO) propose different cut-offs for handgrip strength and muscle quality index/or muscle mass [8,9,10,11,12].

More recent definition criteria proposed by the EWGSOP2-2018 are that sarcopenia is a concomitant occurrence of low muscle quality index/or muscle mass, low muscle strength, and low physical performance [11]. All the three together represent severe sarcopenia, but low muscle strength (a.k.a. dynapenia) is better than muscle mass in predicting adverse outcomes such as all-cause mortality, cardiovascular mortality, and cardiovascular disease [13,14].

Confirming this, Souza et al., using the criteria from the EWGSOP-2010 and FNIH, displayed a prevalence of 11.9 and 28.7%, respectively, in CKD patients. However, no differences between male and females were reported [15]. In light of these findings, the EWGSOP2 guidelines, which were originally designed for older adults, have not been validated for patients with CKD. This raises concerns about their relevance for this population. In CKD, the mechanisms responsible for muscle loss and decreased strength can differ significantly from those associated with aging. Specifically, strength often declines more rapidly than muscle mass, a nuance that these guidelines do not adequately address [16]. Research indicates that the current criteria fail to identify high-risk patients effectively, as individuals with reduced strength (a condition known as dynapenia) often present similar disease stages and clinical profiles to those with normal muscle function [16].

Alarmingly, approximately 18% of pre-dialysis CKD patients are dynapenic, suggesting that many at-risk individuals may go unrecognized under the standard definitions [16]. Moreover, since a decline in muscle strength is more closely linked to poor outcomes in CKD than a loss of muscle mass, relying exclusively on the EWGSOP2 guidelines may result in the oversight of critical prognostic factors necessary for effective patient management [16]. Also, differences between males and females are important to verify, as men display more rapid renal function deterioration than females, and this may potentially affect the prevalence of sarcopenia [17]. In chronic kidney disease, the accumulation of uremic toxins adversely affects Leydig cells, leading to a decrease in testosterone production. Furthermore, the ongoing inflammation associated with the disease can suppress the hypothalamic–pituitary–gonadal axis, resulting in even lower testosterone levels. Additionally, malnutrition, commonly encountered in the advanced stages of kidney failure, impairs the body’s ability to synthesize testosterone [18].

Furthermore, females display a more protective effect than males. Estrogens have been shown to downregulate the expression and activity of the renin–angiotensin system, upregulate the activity of Ang (1–7) and angiotensin-converting enzyme 2 (ACE2), and exert protective effects on preserving beta cell function and preventing apoptosis induced by metabolic injuries [19,20]. Furthermore, it was demonstrated that a low skeletal muscle mass index was associated with an increased prevalence of CKD in males, but a lower prevalence of CKD in females [21]. Thus, the etiology underlying sarcopenia in CKD patients can be complex and sex-specific, but mechanisms underlying in this specific disease have not been investigated yet.

Another critical aspect of CKD patients is the prevalence of obesity. Obesity in non-CKD patients has been reported to negatively affect muscle strength and muscle quality index/or muscle mass [22,23,24,25]. However, traditional metrics, such as body mass index (BMI), may not effectively predict adverse clinical outcomes in patients with CKD compared to waist circumference and waist-to-height ratio. Given the negative association between obesity and kidney outcomes, where glomerular hyperfiltration is driven by changes in growth factors and adipokines that result in fibrosis and glomerulosclerosis, it is important to explore this relationship using different obesity indices, particularly in the context of dynapenia and sarcopenia [24,25,26,27].

Despite the increasing awareness of sarcopenia and dynapenia among patients with CKD, few studies have thoroughly examined their prevalence and how they are associated with obesity, particularly from a sex-specific perspective. The existing literature often does not differentiate between sexes or presents inconsistent findings. Additionally, the relationship between obesity and muscle quality, especially regarding sex differences, is not well understood. This represents a significant gap in knowledge, considering the biological, hormonal, and metabolic differences between male and female patients that may influence muscle wasting patterns and responses to treatment.

To enhance awareness and care for sarcopenia in CKD patients, this study aims to compare the handgrip strength and muscle quality index/or muscle mass of male and females CKD patients with reference values for dynapenia and sarcopenia diagnosis proposed by the EWGSOP2-2018 and FNIH [11,12], as well as to evaluate the correlation between obesity indices with handgrip strength and muscle quality index/or muscle mass.

Our findings may be valuable for sex-specific design development and stratified interventions for sarcopenia, offering attractive insights for practical diagnostics and management attached to musculoskeletal health in CKD. Our study hypothesizes that (1) men display a lower hand grip and muscle quality index/or muscle mass compared to reference values, and (2) obesity indexes negatively affect handgrip strength and muscle quality index/or muscle mass measurements.

## 2. Materials and Methods

### 2.1. Study Population

All participants provided written informed consent, and the study protocol was approved by the Institutional Review Board (protocol number: 23007319.0.0000.0029), in accordance with the Declaration of Helsinki [28]. This analysis is part of a larger project (“Different models of physical training in CKD subjects under conservative and hemodialysis stage”) conducted from September 2021 to December 2024. The current study adopts a cross-sectional, observational design, reported in line with the Strengthening the Reporting of Observational studies in Epidemiology (STROBE) guideline [29].

Out of 180 initially screened CKD patients, 78 individuals undergoing maintenance hemodialysis were included based on predefined criteria: (1) ≥3 months on conventional hemodialysis (≥3 sessions/week) and (2) absence of severe medical events in the previous 3 months. Exclusion criteria included recent cardiovascular events, autoimmune or congenital kidney diseases, orthopedic limitations, severe neuropathy, and unstable heart failure.

The study procedures were conducted over three separate visits, carried out within a pre-planned assessment period lasting one to two weeks per participant. Small groups of patients were enrolled weekly over a six-month period, and their evaluations were distributed across consecutive days to ensure consistency in environmental and logistical conditions, thereby minimizing inter-participant variability. The first visit involved participant screening, including a review of medical history, current medication use, and anthropometric measurements. During the second visit, biochemical analyses were performed for all 78 participants, with venous blood samples collected from the forearm between 7:00 and 9:00 a.m. following a 12 h overnight fast. The third visit consisted of body composition measurements, handgrip strength assessment, and functional performance testing.

### 2.2. Anthropometry and Body Composition

Participants’ body weight was measured using a mechanical scale (Filizola, São Paulo, Brazil), and height was assessed with the integrated stadiometer (precision: 0.5 cm). From these values, body mass index (BMI) was calculated as weight in kilograms divided by height in meters squared. Waist circumference was measured with a non-elastic tape, positioned midway between the iliac crest and the lower rib. Additionally, the waist-to-height ratio was computed. On the day of the dual-energy X-ray absorptiometry (DXA) scan, participants adhered to pre-test guidelines from prior studies, which included refraining from physical activity, observing a four-hour fast, and consuming a light meal without fluids. Upon arrival at the lab, subjects wore lightweight clothing and removed all metal-containing accessories [30]. The DXA scans were conducted by a trained technician using the Prodigy Advance Plus Lunar system (GE Healthcare, Madison, WI, USA), with daily calibration verified using a standard phantom [30]. During scanning, participants lay supine with arms positioned alongside the body, palms facing the thighs, and legs stabilized with non-elastic straps at the knees and ankles. The scan yielded measures of fat mass (FM) and fat-free mass (FFM), all of which were performed and interpreted by the same investigator. The coefficient of variation for both FM and FFM was below 1%.

### 2.3. Biochemical Analysis

Fasting venous blood samples (8–12 h) were collected from the antecubital vein to assess interleukin-6 (IL-6) levels. Participants were instructed to abstain from physical activity for 48 h prior to collection. All samples were centrifuged, aliquoted, and stored at −80 °C until analysis. IL-6 concentrations were measured in triplicate using validated ELISA kits (R&D Systems, Minneapolis, MN, USA), following the manufacturer’s protocols. Blood collection was performed at the Laboratório de Imunogerontologia e Biologia Molecular Aplicada ao Exercício of the Universidade Católica de Brasília, a controlled environment in accordance with WHO guidelines for phlebotomy [31]. Collections occurred between 7:00 and 9:00 a.m., with ambient temperature maintained between 22 °C and 24 °C to ensure sample stability.

### 2.4. Functional Performance Test

Functional mobility was assessed using the Timed Up and Go (TUG) test, following established procedures [32]. After a trial of familiarization, participants were timed while rising from a seated position, walking 3 m, turning around an obstacle, and returning to sit. The best of three trials, separated by 60 s rest, was recorded.

### 2.5. Muscle Quality Index and Handgrip Strength Evaluation

Muscle quality index (MQI) was assessed by adjusting handgrip strength for BMI. Based on FNIH criteria, low muscle mass was defined as a handgrip strength/BMI ratio below 1.0 kg/kg for men and 0.56 kg/kg for women [12]. This method is widely used in large-scale studies due to its practicality and clinical relevance [12,30,33,34,35,36,37,38].

Dynapenia was identified using EWGSOP2-2018 thresholds: <27 kg for men and <16 kg for women, respectively [11].

Handgrip strength was measured with a Jamar^®^ hydraulic dynamometer (Sammons Preston, Bolingbrook, IL, USA), with participants seated, elbows at 90°, forearms neutral, and wrists slightly extended. Three trials were performed on the arm opposite the fistula, with 60 s rest between attempts; the average was used for analysis [39,40].

### 2.6. Statistical Analysis

Data are displayed as mean and standard deviation (SD). Normality was verified by Shapiro–Wilk test, z-score within ± 2.58, and Q-Q plot and boxplot graphics. A one-sample *t*-test was used to compare the mean values of participant handgrip strength and MQI with reference values previously cited in the literature as 27 kg (male cut-off value), 16 kg (female cut-off value), 1.0 kg/kg (male cut-off value), and 0.56 kg/kg (female cut-off value), and Cohen’s d effect sizes of 0.2, 0.5, and 0.8 were considered small, medium, and large, respectively [11,12,41,42]. A Pearson correlation was applied between obesity index (e.g., waist circumference, BMI, waist/height ratio, and percent body fat) with handgrip strength and MQIBMI. Values below 0.2, 0.2–0.49, 0.50–0.80, and >0.80 for strength of association (r) were considered trivial, small, moderate, and large, respectively [42,43]. Also, an independent *t*-test was applied for comparisons of baseline characteristics. A chi-square test (ꭓ^2^) was performed to analyze the baseline characteristics of the participants with regard to medications and diseases. When expected cell frequencies were lower than five and considering that our study displays small datasets and unbalanced tables, Fisher’s Exact test with exact significance was used.

Considering that a deterioration of renal function is more rapid in male than female patients, the power calculation was directed to male participants in this study [17]. Considering a mean difference of 5.0 kg for handgrip strength as the minimal clinically important difference, a population mean of 27 kg, with an alternative hypothesis of 21.27 kg, the standard deviation of the population was estimated at 9.8 for males, with a mean age of 55 years [44,45]. The quantity of participants to detect an effect size of 0.58 (medium), given α = ꞵ = 0.05 (means: difference from constant—one sample case). The results indicate the need of 25 participants to ensure a power of >0.80. For statistical analysis, JASP, GraphPad, and G*Power 3.1.6 software were used. An alpha level of *p* ≤ 0.05 was considered statistically significant for all comparisons [46,47,48].

## 3. Results

No significant differences between males and females were verified for baseline characteristics, as seen in Table 1. However, for dynapenia and sarcopenia, males displayed a significantly higher frequency of 88.6% (relative risk of 5.14 with 95% CI of 2.28–11.60, higher than women) and 75% (relative risk of 3.15 with 95% CI of 1.88–5.30, higher than women), respectively, than females with CKD patients.

For handgrip strength, male participants had a mean of 21.27 ± 5.08 kg, which was significantly lower than the reference value of 27 kg. The mean difference was −5.72 kg (95% CI: −7.27 to −4.18; *p* = 0.001), with a Cohen’s *d* of −1.13 (95% CI: −1.50 to −0.74), indicating a large effect size, as values above 0.8 are considered large according to standard thresholds (0.2 = small; 0.5 = medium; 0.8 = large) (Figure 1). Similarly, for MQI_BMI_, men presented with a mean of 0.80 ± 0.23, which was significantly lower than the reference value of 1.0. The mean difference was −0.19 (95% CI: −0.26 to −0.12; *p* = 0.001), with a Cohen’s *d* of −0.82 (95% CI: −1.16 to −0.48), also reflecting a large effect size (Figure 1).

In contrast, for handgrip strength, female participants showed a mean of 20.88 ± 5.63 kg, significantly higher than the reference value of 16 kg. The mean difference was 4.88 kg (95% CI: 2.92 to 6.85; *p* = 0.001), with a Cohen’s *d* of 0.87 (95% CI: 0.47 to 1.26), also indicating a large effect size (Figure 1). For MQI_BMI_, females had a mean of 0.79 ± 0.24 kg/kg, which was significantly above the reference value of 0.56. The mean difference was 0.23 (95% CI: 0.15 to 0.35; *p* = 0.001), and the Cohen’s *d* was 0.98 (95% CI: 0.56 to 1.38), confirming a large effect as well (Figure 1).

For correlation between obesity indexes with handgrip strength for male participants, a negative significant correlation was observed for waist circumference (*p* = 0.005, low effect size), waist/height ratio (*p* = 0.005, low effect size), and percent body fat (*p* = 0.032, low effect size) (Table 2). Also, a negative significant correlation was observed between MQI_BMI_ and waist circumference (*p* = 0.001, moderate effect size), BMI (*p* = 0.001, moderate effect size), waist/height ratio (*p* = 0.001, moderate effect size), and percent body fat (*p* = 0.001, moderate effect size) for male participants (Table 2).

For correlation between obesity indexes with handgrip strength for female participants, no significant negative correlations were observed for waist circumference (*p* = 0.47, trivial effect size), BMI (*p* = 0.58, trivial effect size) waist/height ratio (*p* = 0.44, trivial effect size), and percent body fat (*p* = 0.25, trivial effect size) (Table 2). Also, a negative significant correlation was observed between MQI_BMI_ and waist circumference (*p* = 0.012, low effect size), BMI (*p* = 0.007, low effect size), waist/height ratio (*p* = 0.011, low effect size), and percent body fat (*p* = 0.001, moderate effect size) for female participants (Table 2).

## 4. Discussion

This study aimed to evaluate sex-specific differences in handgrip strength and MQI among patients with CKD (Figure 2). It compared these values to establish diagnostic cut-offs for sarcopenia and dynapenia while also examining the relationship between obesity indices and muscle function. Consistent with our initial hypothesis, male participants exhibited significantly lower handgrip strength compared to the reference value of 27 kg, with a mean difference of −5.72 kg. Additionally, their mean MQI for BMI was 0.80, which was significantly lower than the reference value of 1.0, with a mean difference of −0.19.

In contrast, female participants did not show a significant deficit when compared to their respective reference cut-offs. Furthermore, obesity indices were negatively correlated with MQI in both sexes. However, an inverse association with handgrip strength was observed only in males. These findings highlight the importance of considering sex-specific patterns in the assessment and management of muscle wasting in CKD, emphasizing the need for tailored diagnostic criteria and interventions.

To our knowledge, the results of the current study are unique compared to other studies [49,50] because we focused our analysis on recent definition criteria by the EWGSOP2-2018, muscle quality (FNIH), and sex differences [11,12].

The first study diagnosed sarcopenia using criteria established by the Asian Working Group for Sarcopenia [9]. It found that sarcopenia is common not only in advanced stages of CKD but also in its earlier stages, highlighting the need for early detection and management. While the first study indicated that men exhibit a higher incidence of sarcopenia compared to women, this difference was not statistically significant due to the small sample sizes involved [49].

The second study diagnosed sarcopenia by combining two criteria: low muscle mass and reduced gait speed [50]. Specifically, low muscle mass was defined as a percentage that falls more than two standard deviations below the mean for young adults of the same sex and ethnic background [51]. Additionally, a low gait speed was identified using a validated measurement for the older adult population, defined as a walking speed of less than 0.8 m/s in the 4 m walking test [52].

This situation clearly demonstrates the inadequacy of these guidelines for the population with CKD. The mechanisms of muscle loss and reduced strength differ significantly from those associated with aging, with strength declining more rapidly than muscle mass, a key distinction overlooked in these guidelines. Research shows that current criteria fail to identify high-risk patients, as those with dynapenia often present with clinical profiles akin to individuals with normal muscle function. Recognizing and addressing these issues is essential for improving patient outcomes [16].

In addition, the focus for sarcopenia in CKD populations should be on the disease-related secondary type, as this form of sarcopenia is induced by chronic illness [4]. This distinction is important because treatment goals are not aimed at restoring mobility and quality of life but rather at reducing mortality rates [4]. Until we gain a better understanding of the appropriate cut-off levels for CKD patients, it is advisable to use the criteria proposed by the EWGSOP2-2018 and FNIH for diagnosing sarcopenia in this population [4,11,12].

In the present study, male CKD patients displayed lower muscle strength (dynapenia) and low muscle mass/or muscle mass index when compared to the reference values proposed by EWGSOP2-2018 and FNIH [11,12]. Also, the prevalence of dynapenia and sarcopenia was higher in male than female CKD patients. This might indicate that gender and potential mechanisms—such as the receptor-mediated effects of sex hormones on glomerular hemodynamics, mesangial cell proliferation, matrix accumulation, and the release of cytokines, vasoactive agents, and growth factors—probably increase the risk of sarcopenia in male CKD patients compared to females [17].

Despite us not evaluating the sex-specific physiological mechanisms underlying sarcopenia and dynapenia, male hypogonadism is a common occurrence and may be further aggravated by other comorbidities associated with CKD [53]. A notable reduction in testosterone levels accompanied by an elevation in estradiol levels among male individuals diagnosed with diabetic nephropathy was discovered [53]. Furthermore, higher serum myostatin levels were associated with muscle atrophy during CKD [54] and sarcopenia prevalence in older men [55], which can contribute to muscle wasting in this specific sex.

Also, although no differences between males and females were verified for IL-6 in blood circulation, males displayed a higher level than females CKD patients (mean difference of 0.32 pg/mL). It is important to note that values higher than 1.80 pg/mL for IL-6 are considered as high proinflammatory status, which negatively affects muscle strength and muscle mass/or muscle quality index [22,56]. Interestingly both groups display high proinflammatory status.

For low muscle strength, males demonstrated a 5.14 times higher relative risk than females of being diagnosed with dynapenia. This is a very important result, as handgrip strength is better than muscle mass in predicting adverse outcomes such as all-cause mortality, cardiovascular mortality, and cardiovascular disease [13,14]. A previous study demonstrated that handgrip strength was an independent predictor of all-cause mortality in maintenance dialysis patients [13]. For males, for example, a cut-off value to predict mortality was ≤22.5 kg [13]. Interestingly and worrisomely, male participants in this study displayed a mean handgrip of 21.27 kg below the cut-off values previously reported and below the values of EWGSOP2-2018, with a large effect size, which communicates the practical significance of the results and the practical consequences of the findings for daily life in patients with CKD [11,42]. Although a lack of consensus of handgrip strength values for patients with CKD exists, further investigation should be reinforced in this important field of research.

As well as low muscle strength, males demonstrated a relative risk 3.15 higher than females of being diagnosed with sarcopenia. By normalizing handgrip strength to BMI, here defined as MQI, a significant explanator of strength not captured within age and gender is accommodated, and 75% of male participants in this study displayed a mean of 0.80 with a significant mean difference, of −0.19 compared with the reference value of 1.0 and with a large effect size [12,42,57]. This is an actual result, as MQI has been recognized as an independent risk factor for overall health and mortality in older adults, and the association of MQI and mortality was more robust for males than females [58]. This result confirms the importance of MQI measurements in CKD patients in future research.

Considering that obesity relates to inflammation in CKD patients, obesity indexes negatively correlated with MQI variables for males and females, and a negative correlation with handgrip strength was verified for male patients with CKD but not females. These results reinforce that obesity should be considered a preventable risk factor in CKD patients, as there exists a negative association between obesity and kidney outcomes, such as glomerular hyperfiltration caused by growth factors and adipokine alterations that lead to fibrosis and glomerulosclerosis [26,27]. Researchers and health professionals should approach the assessment of obesity indices in patients with CKD with caution. While BMI is a commonly used, cost-effective, and practical tool, it does not distinguish between fat mass and lean body mass [24]. In CKD patients, changes in muscle mass can occur due to the disease, potentially leading to misunderstandings regarding the relationship between BMI, muscle strength, and MQI. On the other hand, measurements of waist circumference and waist-to-height ratio are strong predictors of central obesity and are linked to negative clinical outcomes and increased mortality among CKD patients [24,25]. These measurements may provide a better correlation with muscle strength and MQI parameters.

## 5. Limitations

This study has several methodological limitations. Firstly, the sample size was small, and we did not assess clinical and longitudinal changes in dynapenia and sarcopenia following a single hemodialysis session. Secondly, this study’s cross-sectional design restricts our ability to draw causal conclusions between the variables and outcomes. Thirdly, managing medications in patients with chronic kidney disease (CKD) is challenging due to impaired kidney function affecting drug metabolism and elimination. Many CKD patients also have hypertension and diabetes, increasing their risk of complications that can impact treatment efficacy, safety, muscle strength, and overall quality of life [59]. A major concern is the higher risk of adverse drug reactions. For instance, nonsteroidal anti-inflammatory drugs (NSAIDs) can worsen CKD by causing fluid retention and acute kidney injury (AKI) [59].

Similarly, while renin–angiotensin–aldosterone system blockers are beneficial for cardiovascular health, their use with NSAIDs or diuretics can lead to hypotension and AKI. Also, proton pump inhibitors (PPIs) have been linked to progressive renal impairment when used long-term [59]. In addition, drug accumulation and toxicity are critical issues, as altered pharmacokinetics can result in slower metabolism and excretion of medications in CKD patients. Inadequate dosing of drugs like metformin may cause lactic acidosis, while sulfonylureas can lead to hypoglycemia due to prolonged effects [59]. Additionally, polypharmacy is a pressing concern, increasing the likelihood of drug–drug interactions and inappropriate use [59]. Given these complexities, interprofessional collaboration, particularly with clinical pharmacists, is essential. Their expertise in identifying potential drug-related issues, reviewing dosages, and supporting clinical decisions can significantly enhance patient safety and treatment quality.

Additionally, we did not conduct a formal power calculation for our secondary objective, which aimed to examine the association between obesity indices and muscle parameters. Sub-analyses by sex may also be limited due to smaller group sizes, which could affect the detection of more subtle associations. Furthermore, we were unable to use logistic regression or ridge regression analysis to predict the categorical outcome of gender based on obesity indices, as this method does not address the issue of multicollinearity. Lastly, there is no consensus on the operational definition of sarcopenia in patients with chronic kidney disease (CKD), and using cut-off values established for the general population may not be appropriate for this specific group [4,60].

## 6. Conclusions

In summary, males with CKD display lower muscle strength and MQI values compared to reference values proposed by EWGSOP2-2018 and FNIH [11,12] and demonstrate a superior relative risk of being diagnosed with dynapenia and sarcopenia than women. Furthermore, higher obesity indexes negatively impact muscle strength and mass in CKD patients, with notable sex-specific differences in the prevalence of dynapenia and sarcopenia. While our study shows a higher relative risk for these conditions in males, the biological mechanisms such as hormonal regulation and inflammation require further exploration, particularly regarding testosterone deficiency and myostatin activity.

Additionally, the negative correlation between obesity indices and muscle quality, especially in males, suggests that waist circumference and waist-to-height ratio may be more relevant for intervention than BMI, which often misclassifies CKD patients. Thus, future research should validate these measures as potential screening tools for sarcopenia. Also, studies should investigate MQI’s prognostic value for clinical outcomes in CKD. Finally, given the absence of consensus on cut-off values for sarcopenia and dynapenia in CKD, there is a critical need for CKD-specific diagnostic thresholds developed from diverse patient populations to enhance diagnostic precision and inform targeted interventions.

## Figures and Tables

**Figure 1 healthcare-13-01621-f001:**
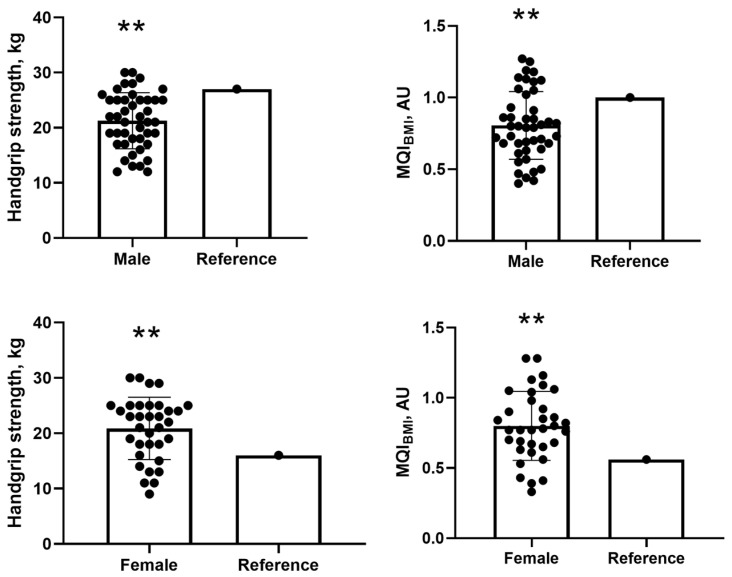
Comparisons of handgrip strength and MQI with cut-off values for male and female participants. Note: MQI = muscle quality index. ** One-sample *t*-test at the 0.01 level (2-tailed). Values presented as mean and standard deviation (SD).

**Figure 2 healthcare-13-01621-f002:**
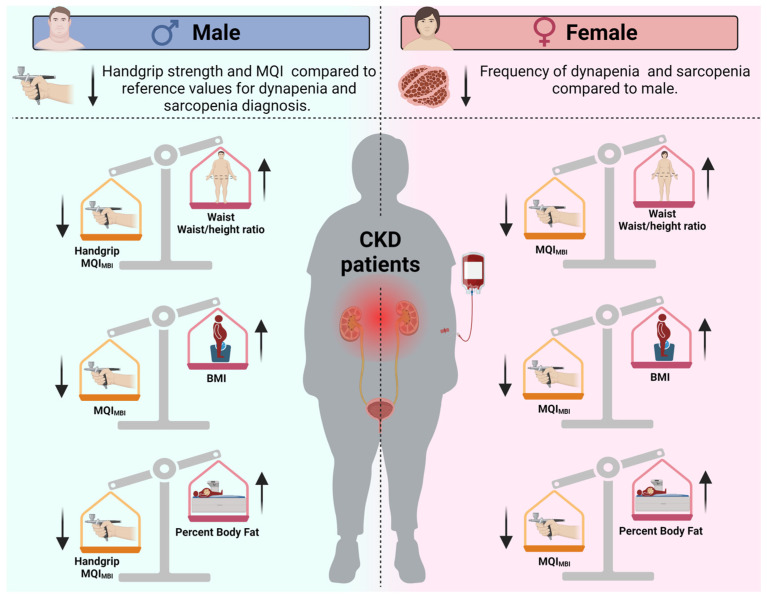
Overview of sex-based differences in sarcopenia and dynapenia prevalence, as well as the effects of obesity indexes on muscle strength and muscle quality in chronic kidney disease patients. The arrows indicate whether the values of the indicators are increased (↑) or decreased (↓) in patients with chronic kidney disease according to sex. Note: BMI = Body mass index; MQI = muscle quality index.

**Table 1 healthcare-13-01621-t001:** Basal characteristics of participants.

Variables		Male (*n* = 44)	Female (*n* = 34)	*p*
Age, years		57.77 ± 4.28	57.26 ± 3.79	0.58
Body weight, kg		73.26 ± 14.23	70.96 ± 13.69	0.47
Height, m		1.64 ± 0.08	1.62 ± 0.07	0.40
BMI, kg/m^2^		26.89 ± 3.01	26.52 ± 2.95	0.59
Waist circumference, cm		96.55 ± 13.04	94.29 ± 11.39	0.42
Waist/height ratio		0.58 ± 0.05	0.57 ± 0.04	0.46
Body fat, kg		29.83 ± 8.78	28.62 ± 8.51	0.54
Body fat, %		39.93 ± 4.47	39.55 ± 4.46	0.71
Fat-free-mass, kg		43.43 ± 5.70	42.33 ± 5.35	0.40
Handgrip strength, kg		21.27 ± 5.08	20.88 ± 5.63	0.74
MQI_Body Weight_, kg/kg		0.30 ± 0.10	0.30 ± 0.10	0.99
MQI_BMI_, AU		0.80 ± 0.23	0.79 ± 0.24	0.89
TUG, seconds		17.53 ± 3.78	17.13 ± 3.90	0.64
IL-6, pg/mL		3.67 ± 0.92	3.34 ± 0.82	0.10
Time of hemodialysis, weeks		54.86 ± 9.71	55.82 ± 9.08	0.75
Diseases				ꭓ^2^
Hypertension, *n* (%)	Yes	44 (100%)	34 (100%)	1.00
	No	0	0	
Diabetes, *n* (%)	Yes	26 (59.1%)	18 (40.9%)	0.64
	No	18 (40.9%)	16 (47.1%)	
Medications				
Erythropoietin, *n* (%)	Yes	30 (68.2%)	30 (88.2%)	0.057
	No	14 (31.8%)	4 (11.8%)	
Statins, *n* (%)	Yes	30 (68.2%)	29 (85.3%)	0.11
	No	14 (31.8%)	5 (14.7%)	
ACEi, *n* (%)	Yes	35 (79.5%)	30 (88.2%)	0.37
	No	9 (20.5%)	4 (11.8%)	
Cut-off values				
Dynapenia, *n* (%)	Yes	39 (88.6%)	8 (23.5%)	0.001
	No	5 (11.4%)	26 (76.5%)	
Sarcopenia, *n* (%)	Yes	33 (75.0%)	5 (14.7%)	0.001
	No	11 (25.0%)	29 (17.4%)	

Note: Data are shown as mean ± SD. Independent *t*-test was used to analyze the baseline characteristics of participants. BMI = body mass index; MQI = muscle quality index; AU = arbitrary unit; TUG = Timed Up and Go test; IL-6 = interleukin-6; ACEi = angiotensin-converting enzyme inhibitors.

**Table 2 healthcare-13-01621-t002:** Correlation between obesity indexes with handgrip and MQI for male and female participants.

Male (*n* = 44)				
Handgrip	Waist	BMI	Waist/height ratio	Percent body fat
1	−0.41 **	−0.23	−0.41 **	−0.32 *
MQI_BMI_	Waist	BMI	Waist/heigh ratio	Percent body fat
1	−0.65 **	−0.59 **	−0.66 **	−0.65 **
**Female (*n* = 34)**				
Handgrip	Waist	BMI	Waist/height ratio	Percent body fat
1	−0.12	−0.09	−0.13	−0.20
MQI_BMI_	Waist	BMI	Waist/height ratio	Percent body fat
1	−0.42 *	−0.45 **	−0.43 *	−0.52 **

Note: BMI = body mass index: MQI = muscle quality index. ** Correlation is significant at the 0.01 level (2-tailed). * Correlation is significant at the 0.05 level (2-tailed).

## Data Availability

The data that support the findings of this study are available from the corresponding author (Jonato Prestes) upon reasonable request.

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
