# Peer review of "Impact of Sarcopenia, Dynapenia, and Obesity on Muscle Strength and Quality in Chronic Kidney Disease Patients: A Sex-Specific Study"

_healthcare, 2025, doi:10.3390/healthcare13131621_

Round 1
Reviewer 1 Report
Comments and Suggestions for Authors
First of all, I would like to thank you for invited to read the document.
Each of the comments shared are intended to improve the study.
The comments can be found in the PDF document.
Also, some of the comments on some of the points that need to be reworded in the paper are shared below:
Title
We suggest revising the title and adjusting it to the journal's guidelines.
Sex-Specific Prevalence of Sarcopena, Dynapenia and Negative Effects of Obesity Indexes on Muscle Strength and Muscle Quality in Chronic Kidney Disease Patients
It is also suggested that the title be revised by placing the variables in the first part and the population characteristics at the end. This will help to understand more clearly the object of study.
Abstract
It is necessary to refer to it at the beginning of the abstract.
In this section it would be important to refer to the particular findings found in this research.
Mention what was the impact of sex on the prevelance of sarcopenia and dinapenia?
Introduction
What does the scientific evidence say about this? Some characteristics of the interventions can be referred to.
It is suggested to review whether it is necessary to add a paragraph before mentioning the objectives on the lack of studies analyzing the sex-specific prevalence of sarcopenia, dinapenia and negative effects of obesity rates on strength and muscle quality in patients with chronic kidney disease. This would give more foundation to the study presented.
Material and method
Study population
It is suggested that the study design be detailed.
The question arises whether a power test was used to define the sample size.
In how many days were all the evaluations performed. The duration of each of the tests used is not precisely detailed. It would be necessary to be able to explain it more precisely.
Biochemical analysis
It is suggested to be able to add information on the place where the blood samples were collected and to detail the ambient temperature.
Also, were all the blood samples taken from the 78 participants taken on the same day?
Results
Table 1. Two things are suggested:
First, add this table of sample characteristics in the method section.
Second, add a table with all descriptive and inferential statistics reporting significance values and effect sizes for each of the variables in response to sex.
When reading the results, some variables are referred to as significant, but it is not clear where they came from. Adding the table helps to understand all the results of the study.
Discussion
It is necessary to review the discussion, there are being added results of variables that were not analyzed in the present study and therefore the results found in the biochemical variables, functional performance test and evaluation of muscle quality index and hand grip strength are not discussed.
It is necessary to restructure.
We suggest starting the discussion with the objective of the present study and then detailing the main findings in response to sex.
What types of results are being compared or reported in these studies [45,46] so that we know that the results of the present investigation are unique.
Conclusions
It is necessary to add two subsections, one focused on the limitations of the study and the other on the future perspectives of the research. This would help to further strengthen the presented study and shed light on future research.
Finally, I thank the authors for the excellent work and encourage you to review the comments shared.

Reviewer 2 Report
Comments and Suggestions for Authors
The manuscript titled "Sex-specific prevalence of sarcopenia, dynapenia and negative effects of obesity indexes on muscle strength and muscle quality in chronic kidney disease patients" presents a significant area of inquiry. However, identifying key gaps and weaknesses is crucial to understanding its robustness and contributions. Here’s an overview of the comments and questions derived from the review:
-
Introduction and Background:
- Gap: While the introduction posits the necessity to explore sex differences in CKD, it does not sufficiently detail how this study fills a critical gap in the existing literature concerning varying diagnostic criteria for sarcopenia and dynapenia.
- Question: How does the lack of standardized criteria for these conditions potentially influence the findings or their interpretation?
-
Methodology:
- Weakness: The small sample size (N=78) raises concerns about statistical power and reliability of the findings.
- Question: What measures were taken to ensure that the sample size was sufficient to detect meaningful differences, especially in a sub-analysis by sex?
-
Data Analysis:
- Concern: The reliance on specific obesity indices without discussing alternative measurements that could yield different results is noted.
- Question: Why were certain obesity indices chosen over others, and could other measures present additional insights into the relationships observed?
-
Discussion:
- Weakness: The discussion does not sufficiently address conflicting studies, nor does it present a critical examination of the implications of these discrepancies.
- Question: What are the reasons for the lack of alignment with previous research regarding obesity indexes' impact on muscle strength in CKD patients?
-
Conclusion:
- Gap: Relays findings without linking them back to specific clinical implications or future research directions.
- Question: What practical recommendations can clinicians derive from the findings, and how might these impact clinical practices for CKD management?
-
Future Directions:
- Weakness: Limited exploration of how these findings can inform larger-scale studies or interventions is present.
- Question: What would be the next steps in studying this topic, particularly considering the need for longitudinal data?
Reviewer 3 Report
Comments and Suggestions for Authors
Dear authors,
Congratulation on your important work. This study aimed to compare the handgrip strength and MQI of male and female chronic kidney disease (CKD) undergoing hemodialysis, referencing diagnostic criteria for dynapenia and sarcopenia, and to evaluate the correlation between obesity indices and these muscle-related parameters. Here are some of my concerns to share.
- Like you mentioned, this is a cross-sectional study with limited participants, did you adjust for their different medication history? As far as I know, patients during hemodialysis might under certain treatment for CKD. Furthermore, from your table 1, it seems that most participants were diagnosed with CVD, e.g. hypertension, diabetes, etc. Those patients might under certain treatment for different metabolic syndromes. Therefore, these could be both intriguing and confounding factors which can affect the MQI and handgrip strength.
- Like I mentioned above, you may want to discuss the pharmaceutical targets and underlying mechanisms of CKD treatment and most commonly adapted medications and associated complications.
- Since the title is regarding sex-specific sarcopenia, obesity on muscle strength of CKD patients under hemodialysis, and the MQI results were normalized to BMI, people may propose the hypothesis that male patients can perform better handgrip strength exercise test than female CKD patients. I am a little bit concerned about your hypothesis intuitively.
- There might be some upcoming research regarding CKD patients medication, surgical procedures and exercise intervention in the last two years since 2022, the current references are only up to the year of 2022, you may want to expand the background section and discussion section with most recent studies.
Overall, this is a well-designed study with good controls. I acknowledge that there are many confounding factors in clinical trials that may not be able to be adjusted to, but I would be happy to recommend this study to be accepted after a minor revision.
Best regards,
Round 2
Reviewer 1 Report
Comments and Suggestions for Authors
Thank you for the opportunity to review the manuscript.
The authors have responded to each of the comments made. However, we suggest reviewing the comments attached to the document, which are also shared below:
Abstract We suggest revising the abstract.
It is too short and lacks depth.
We suggest adding characteristics of the population evaluated and key characteristics of the study.
Keywords
We suggest reviewing and adding other keywords that represent the study.
Discussion
Lines 382-404. We suggest adding a sub-section to address the limitations separately, in order to better understand the difficulties and possible biases of the study.
Lines 405-413. In addition to discussing the limitations of the study, it is suggested that future research prospects be mentioned.
Beyond exposing the limitations of the study, it is suggested to mention future research prospects.
There, the opportunity to develop longitudinal and evaluation studies in other samples can be mentioned.
I can suggest the suitability of the manuscript for publication, once the latest suggestions attached to the document have been corrected.

Author Response
Dear Editor of HealthCare from MDPI
Thank you for the opportunity to resubmit our manuscript. The comments from the reviewers were highly insightful and enabled us to improve the quality of our manuscript. We have made the changes according to their advice. A point-by-point response to the reviewers, as well as the marked-up version (highlighted in red) of the manuscript, are attached.
I believe that we have addressed all questions raised by the reviewers, and the manuscript has greatly improved. Thank you again for your time and effort in considering this manuscript for publication.
Yours sincerely,
PhD. Jonato Prestes
Thank you for the opportunity to review the manuscript.
The authors have responded to each of the comments made. However, we suggest reviewing the comments attached to the document, which are also shared below:
Abstract We suggest revising the abstract.
It is too short and lacks depth.
We suggest adding characteristics of the population evaluated and key characteristics of the study.
Response: Thank you very much for the important suggestions. We changed the abstract as requested. Please refer to lines 26 to 44.
Keywords
We suggest reviewing and adding other keywords that represent the study.
Response: Thank you very much for the important suggestions. We included two more keywords as requested. Please refer to line 45.
Discussion
Lines 382-404. We suggest adding a sub-section to address the limitations separately, in order to better understand the difficulties and possible biases of the study.
Response: Thank you very much for the important suggestions. We added a sub-section as requested. Please refer to line 45.
Lines 405-413. In addition to discussing the limitations of the study, it is suggested that future research prospects be mentioned.
Beyond exposing the limitations of the study, it is suggested to mention future research prospects.
There, the opportunity to develop longitudinal and evaluation studies in other samples can be mentioned.
Response: Thank you very much for the important suggestions. We mentioned the future research prospects as requested in the conclusion section. Please refer to lines 426 to 437.
I can suggest the suitability of the manuscript for publication, once the latest suggestions attached to the document have been corrected.
Reviewer 2 Report
Comments and Suggestions for Authors
The author responded to my question.
Author Response
Dear Editor of HealthCare from MDPI
Thank you for the opportunity to resubmit our manuscript. The comments from the reviewers were highly insightful and enabled us to improve the quality of our manuscript. We have made the changes according to their advice. A point-by-point response to the reviewers, as well as the marked-up version (highlighted in red) of the manuscript, are attached.
I believe that we have addressed all questions raised by the reviewers, and the manuscript has greatly improved. Thank you again for your time and effort in considering this manuscript for publication.
Yours sincerely,
PhD. Jonato Prestes
Thank you for your valuable observations. The quality of the paper substantially improved with your review. Thank you for your effort in helping us identify gaps and enhance our work.